# DEEAPR: Controllable Depth Enhancement via Adaptive Parametric Feature Rotation

## Abstract

Understanding depth of an image provides viewers with a better interpretation of the 3D structures within an image. Photographers utilize numerous factors that can affect depth perception to aesthetically improve a scene. Unfortunately, controlling depth perception *after* the image has been captured is a difficult process as it requires accurate and explicit depth information. Also, defining a quantitative metric of a subjective quality (i.e., depth perception) is difficult which makes supervised learning a great challenge. To this end, we propose DEpth Enhancement via Adaptive Parametric feature Rotation (DEEAPR), which modulates the perceptual depth of an input scene using a single control parameter without the need for explicit depth information. We first embed content-independent depth perception of a scene by visual representation learning. Then, we train the controllable depth enhancer network with a novel modulator, parametric feature rotation block (PFRB), that allows for continuous modulation of a representative feature. We demonstrate the effectiveness of our proposed approach by verifying each component through an ablation study and comparison to other controllable methods.

## 1 Introduction

Enhancing depth perception in 2D images exhibits a more realistic image content to the viewers as it allows for better interpretation of the 3D scene structure. The human visual system uses a variety of cues to inference depth information, such as disparity that arises from binocular vision or motion (Kim et al., 2016). Depth cues in single static images, such as occlusion, shading, or blur, are referred to as pictorial depth cues (O'Shea et al., 1997). Many artists and photographers utilize pictorial depth cues by adding synthetic effects to increase impression of depth in still images. For example, amplifying the defocus blur triggers the depth-of-focus cue, where the objects in the range of focus appear sharp and those further away appear blurry. However, enhancing depth perception of an image by manipulating depth cues *without* explicit depth information of a scene is a challenging task. Moreover, depth perception of a scene is a subjective quality that varies from image to image, which is difficult to learn in a supervised manner.

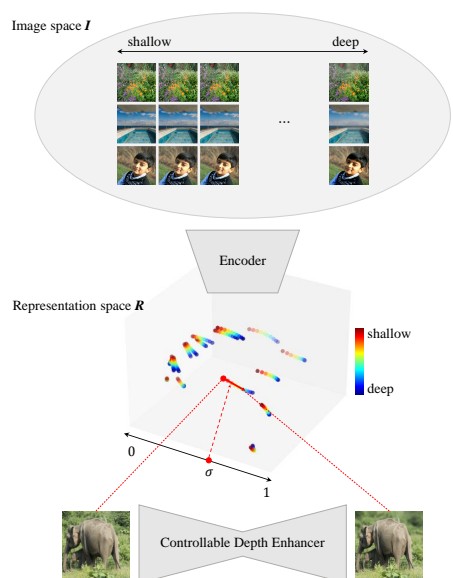

Figure 1: Illustration of DEEAPR framework.

Our final goal is to modulate the perceptual depth of an input scene using a single control parameter without the need for explicit depth information. While the traits of depth perception make supervised training very difficult, we rely on the recent success of unsupervised visual representation learning that considers a high dimensional space in which the degree of similarity is inversely correlated with the distance between instances. As the first step, we embed content-independent depth perception of a scene onto the representation space

by combining contrastive learning (Hadsell et al., 2006) and metric learning (Balntas et al., 2016). The representation space of the encoder then bridges the image space and the control parameter axis. Instead of relying on full supervision, we aim to train the *direction* of depth enhancement during depth enhancer training. Fig. 1 illustrates our DEEAPR framework combining visual representation learning and a controllable neural network. We first train a depth representation encoder in an unsupervised manner, then train the *direction* of depth perception enhancer such that appropriate change would be induced in the image domain when the depth representative feature is modulated.

Our contributions are three-fold. First, we propose a novel strategy to learn visual representation space of style disentangled from image contents. Second, we present a controllable neural network that enhances the depth perception of an input image according to a single control parameter. Lastly, we present a novel modulator, parametric feature rotation block (PFRB), that allows for continuous modulation of feature representation while preserving its norm.

## 2 RELATED WORKS

### 2.1 VISUAL REPRESENTATION LEARNING

Recent success of unsupervised representation learning in natural language processing has motivated the computer vision community to pursue learning a manifold of natural images. To learn an effective feature space for downstream tasks, many works utilized contrastive loss (Hadsell et al., 2006) to discriminate highly correlated instances (i.e., positive pairs) from others. In general, various content-preserving data augmentation is applied to the same image independently in order to generate positive pairs (Chen et al., 2020a). All other examples obtained from different images are considered as negative samples, and the objective is to minimize the distance between positive pairs with respect to myriads of negative samples. Momentum Contrast (MoCo) (He et al., 2020b) is a representative approach which aims to build a dynamic dictionary encoded by a moving-averaged momentum encoder. Furthermore, DASR (Wang et al., 2021) adapts MoCo to learn a content-independent degradation representation for the blind super-resolution task. Our strategy is similar to MoCo v2 (Chen et al., 2020b) and partly motivated from DASR in that we aim to train *content-independent* representation of depth perception via contrastive learning. We use triplet margin loss (Balntas et al., 2016) as an additional regularizer to disentangle depth perception from the image content.

### 2.2 DEPTH PERCEPTION ENHANCEMENT

Interpretation of 3D scene structure from single static images can be obtained from various pictorial depth cues, such as occlusion, relative size, linear perspective, aerial perspective, texture gradients, depth-of-focus and shading (Cipiloglu et al., 2010). These depth cues are utilized in conventional methods by for example, enhancing local contrast (Ritschel et al., 2008; Lee et al., 2014), slightly darkening or lightening the background or partly occluded objects (Luft et al., 2004), or applying shading and/or shadows (Vergne et al., 2011; Lopez-Moreno et al., 2011). While these methods require *explicit* use of depth information at the pixel level, Hel-Or et al. (2017) extract reflectance derivatives of an image to manipulate shading cues. Conventional methods often require precise manipulation of multitude of parameters for plausible and visually pleasing output. More recently, there have been great interest in developing deep learning-based approaches to render artificial depth-of-field effect, often referred to as the Bokeh effect. Indeed, through the AIM Boken Challenges (Ignatov et al., 2019; 2020b), numerous approaches created depth-dependent blur effects by using depth information explicitly (Peng et al., 2022) or implicitly (Ignatov et al., 2020a; Qian et al., 2020; Dutta et al., 2021). Our DEEAPR differs from aforementioned approaches in two major aspects: DEEAPR procudes a depth enhanced image based on a *single* parameter without *explicit* depth information.

### 2.3 CONTROLLABLE NEURAL NETWORK

Conventional deep learning methods learn a *deterministic mapping* for each task and output a *single* image given an input. Controllable methods, on the other hand, provide flexibility in producing images at *continuous* levels for modulation over diverse imagery effects. Controllability has appeared in several low-level vision tasks such as style transfer to permit smooth interpolation between different artistic styles (Dumoulin et al., 2016; Ghiasi et al., 2017; Huang & Belongie, 2017; Wang et al.,

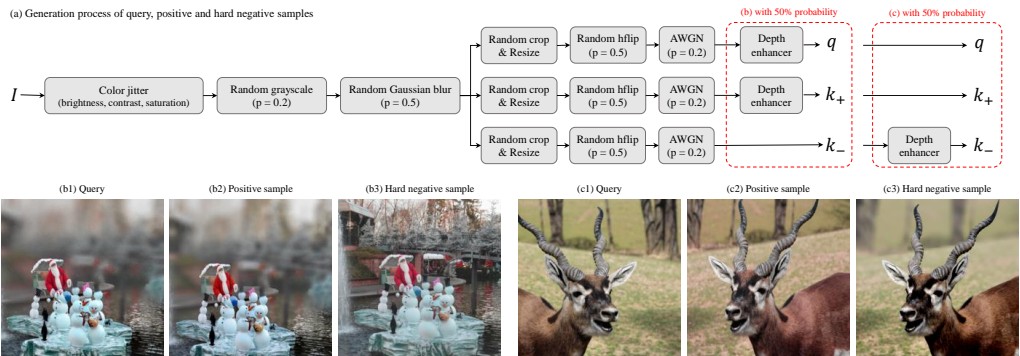

Figure 2: (a) Generation process of query, positive and hard negative sample. (b), (c) The three samples have same image content but the hard negative sample (b3/c3) has different depth perception from the query (b1/c1) and positive sample (b2/c2).

2019b) and image restoration to restore images at continuous levels of degradation (Fan et al., 2019; He et al., 2019; 2020a; Kim et al., 2021; Wang et al., 2019a;b). Various designs of controllable neural networks have been explored, which include tuning different parameters of normalization layers (Dumoulin et al., 2016; Ghiasi et al., 2017; Huang & Belongie, 2017), applying linear interpolation on parameters of multiple correlated networks (Wang et al., 2019b), modifying intermediate layer features (He et al., 2019) and using a network consisting of several fully-connected layers to learn the channel importance for a specified task (He et al., 2020a; Kim et al., 2021). Similarly, Wang et al. (2019a) designed a neural network consisting of a main branch and a tuning branch to modulate the feature of the main branch. While aforementioned approaches applied the control parameter directly to modify the base image generation network, the control parameter in DEEAPR is encoded to modulate the depth representative feature to separate the base network from the variable part, lifting the burden of multitask learning from the base network.

## 3 METHODS

The DEEAPR network consists of *a depth representation encoder* and *a controllable depth enhancer*. The representation encoder is responsible for generating the depth representative *feature* of the input image and the enhancer for generating a depth enhanced *image*. The representation encoder and the enhancer are trained in two separate stages, which are described in Sec. 3.1 and 3.2, resp.

### 3.1 LEARNING DEPTH REPRESENTATION

It is crucial that the content of an image is preserved in the process of depth enhancement. Thus, our goal is to learn a feature representative of depth that can be modulated without affecting the contents of a given image. Due to the lack of a quantitative metric that measures the perception of depth, we devise a *data-driven* approach to extract a feature *representative* of depth via visual representation learning. As current existing contrastive loss-based algorithms are limited in distinguishing content from style (or depth perception in the context of depth enhancement), we introduce a hard negative sample, $k_-$, for each query. The hard negative sample shares the same content as the query and the positive sample, but has different style to reinforce the encoder to embed visual instances according to their style, not content. This is achieved by adding the triplet margin loss (Balntas et al., 2016) as regularization, $\mathcal{L}_{reg}$, to the MoCo v2's framework, as follows:

$$\mathcal{L}_{total} = \mathcal{L}_c + \lambda\mathcal{L}_{reg} = -\log\frac{\exp\left(q \cdot k_+/\tau\right)}{\Sigma_{i=1}^{K}\exp\left(q \cdot k_i/\tau\right)} + \lambda \cdot \max\{(q \cdot k_-) - (q \cdot k_+) + m, 0\}, \quad (1)$$

where $K$ is the dictionary size, $\tau$ is the softmax temperature, $\lambda$ is a regularization parameter and $m$ is a margin. Margin $m$ regulates the difference of correlations between query-positive and query-hard negative pairs. The $\mathcal{L}_{reg}$ term is crucial for disentangling style from content as it forces the query to push away image patches with same content but different depth perception. We analyze the effect of the regularizer in representation space in Sec. 5.1.

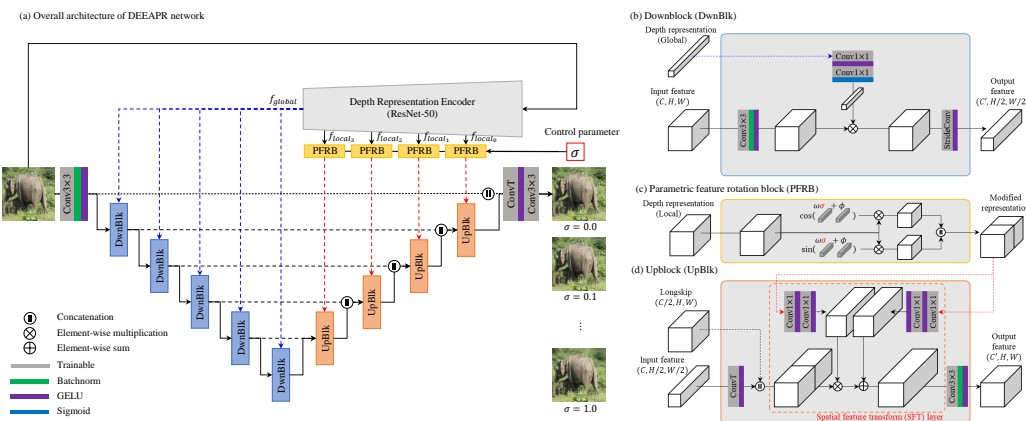

Figure 3: (a) The overall architecture of DEEAPR network and (b), (c), (d) its constituents.

To obtain positive and hard negatives for a given query, we applied pre- and post-transforms depending on their ability to affect the perception of depth. That is, as blur, brightness, contrast and saturation can affect depth perception, we applied these operations commonly across the image to obtain the query, positive and hard negative samples. Crop, resize, horizontal flip, additive white Gaussian noise and hue, on the other hand, have no effect on depth perception, and thus were applied independently. We applied the synthetic depth enhancement model (described in Sec. 4.1) to the query and positive pair or to the hard negative sample with equal probability. We provide concrete examples of query, positive and hard negative samples in Fig. 2(b) and (c).

## 3.2 Learning Depth Enhancement Effect

**Network architecture.** Fig. 3 illustrates the architecture of the DEEAPR network and its constituents. The parametric feature rotation blocks (PFRBs) modulate the depth representative feature according to $\sigma$ and provide modified local representations to the depth enhancer. The depth enhancer, which is responsible for enhancing the depth perception to a given extent, takes on an architectural structure of U-Net consisting of $N = 5$ Downblocks and $N - 1$ Upblocks. The features are downscaled via strided convolution and upscaled via transposed convolution. As the features from the Downblock and Upblock correspond to blurred features of the input due to the down- and upscaling operations, we ensure that the two blocks remain connected via the skip connection to retain sharp as well as contrast and brightness information.

**Downblock and Upblock.** Both the Downblock and the Upblock reference depth representation, but treat the information in different ways. The Downblock references the *global* depth representative feature and applies channel attention to its input to selectively downsample features-of-interest. Each Upblock receives two sets of features: one from the previous Upblock (or the last Downblock in case of the first Upblock) and another from the skip-connection. We ensure that spatial variance is maintained while combining these features by applying spatially varying scales and biases using a spatial feature transform (SFT) layer (Wang et al., 2018). The SFT layer takes the modified *local* depth representation corresponding to each resolution as a condition.

**Parametric Feature Rotation Block (PFRB).** The local depth representation is modulated by hyperparameter, $\sigma$, which controls the strength of the depth enhancement effect. As representative features reside along the unit hypersphere due to the normalization layer, the features are controlled by applying sinusoidal operations on the encoded control parameter. That is, the local representation is multiplied by cosine and sine of $\omega\sigma + \phi$, where $\omega$ and $\phi$ are trainable vectors. Formally,

$$PFRB(f_{local_i}, \sigma) = [\cos(\omega\sigma + \phi) \odot f_{local_i} : \sin(\omega\sigma + \phi) \odot f_{local_i}], \qquad (2)$$

where $\odot$ denotes element-wise multiplication (i.e., Hadamard product) and : denotes concatenation. Notice that PFRB preserves the norm of output features regardless of the value of $\sigma$ by the Pythagorean trigonometric identity. This ensures that the output feature of PFRB has a local norm of 1, which in turn stabilizes training of the depth enhancer module.

**Self-Supervised Training.** Suppose $\sigma \in \mathbb{R}[0,1]$ denotes the control parameter of depth enhancement, where $\sigma = 0$ implies that the target image is the input image and $\sigma = 1$ implies that the target is its depth-enhanced version. Training a depth enhancement network at continuous levels of $\sigma \in \mathbb{R}[0,1]$ in a fully supervised way would require pairs of an input image and a depth-enhanced image at various levels of $\sigma$. However, the initial depth perception of an input differs from image to image, which leads to different perception of depth enhancement despite using the same parameters. Therefore, we utilize the fact that the direction of change remains consistent (i.e., as $\sigma \to 1$, the target image should be perceived deeper) and only aim to train the *direction* of depth enhancement.

Indeed, when training the DEEAPR network, we sample the control parameter $\sigma$ from $\{0,1\}$ with equal probability but aim to infer depth enhanced images in between $\mathbb{R}[0,1]$. Given an image $\mathbf{I}$ and its depth-enhanced version $\mathbf{O}$, the depth enhancer network is trained using L1 loss as a reconstruction loss, where the target is the input $\mathbf{I}$ when $\sigma = 0$ and the target is $\mathbf{O}$ otherwise. Inferencing depth enhanced images at continuous levels of $\sigma \in \mathbb{R}[0,1]$ is made possible despite training at discrete levels of $\sigma \in \{0,1\}$ as the the sinusoidal functions in the PFRB allow for smooth interpolation between the depth representation features that reside on the unit hypersphere.

## 4 Experimental Setup

### 4.1 Modeling Depth Enhancement Effect

We considered two processes to synthesize depth enhanced images: contrast manipulation and circular blur effect. Increasing the contrast of near objects enhances volumetric perception, while decreasing the contrast of the background simulates an atmospheric obscuring effect. To model the depth-dependent contrast effect, we modified the standard haze model (Yang & Sun, 2018):

$$O_{contrast}(\mathbf{x}) = I(\mathbf{x})T(\mathbf{x}) + A(1 - T(\mathbf{x})) \text{ and } T(\mathbf{x}) = \max\{1, \exp(-\eta(d(\mathbf{x}) - \delta_{contrast}))\}, \quad (3)$$

where $I(\mathbf{x})$ is the intensity of scene radiance at position $\mathbf{x}$, $A$ the atmospheric light and $d(\mathbf{x}) \in [0,1]$ the normalized scene depth. The standard haze model only simulates the obscuring effect by setting $A$ to some constant value. However, we observed that this model does not account for the contrast enhancement effect of close objects and creates visually unpleasing artifacts in images with low brightness. Correspondingly, we redefined $A = \bar{I} + \alpha$, where $\bar{I} = \frac{1}{|I(\mathbf{x})|}\sum_{\forall \mathbf{x}} I(\mathbf{x})$ is a global average of the scene radiance and $\alpha$ is a constant. In addition, we did not limit the max value of $T(\mathbf{x})$ to 1 to enhance contrast of relatively near objects (i.e., $d(\mathbf{x}) < \delta_{contrast}$).

Synthetic blur is frequently used in photography to manipulate depth perception of a scene. Images captured with shallow depth-of-field have blurred background, making the in-focus objects stand out. We modified the Bokeh effect synthesis in Wadhwa et al. (2018) and applied a circular blur kernel with its size $r$ linearly increasing with respect to depth, $d(\mathbf{x})$. The blur kernel is applied to regions farther away than the distance $\delta_{blur}$ (i.e., $d(\mathbf{x}) > \delta_{blur}$) and the size of the kernel increases with slope $s$. When applying the kernel, we ignored pixel values whose depth difference exceeded threshold $d_{\max} = 0.1$ with respect to the center pixel, $\mathbf{x}_i$. In summary, the final output image $O(\mathbf{x})$ was obtained by

$$O(\mathbf{x}) = \frac{1}{|\{x_i \in \Omega(\mathbf{x})| \, |d(\mathbf{x}_i) - d(\mathbf{x})| < d_{\max}\}|} \Sigma_{\mathbf{x}_i \in \Omega(\mathbf{x})} \mathbf{1}_{|d(\mathbf{x}_i) - d(\mathbf{x})| < d_{\max}} O_{contrast}(\mathbf{x}_i), \quad (4)$$

where $\Omega(\mathbf{x})$ is a local circular window with radius $r(\mathbf{x}) = \max\{s(d(\mathbf{x}) - \delta_{blur}), 0\}$.

Both effects require accurate depth information of a scene, $d(\mathbf{x})$. For this purpose, we used pretrained DPT (Ranftl et al., 2021) for monocular depth estimation and refined the predicted depth using the fast bilateral solver (Barron & Poole, 2016). We observed that depth prediction by DPT was unreliable in regions where depth changed drastically or absolute depth value was large; thus, we integrated these two heuristics into generating a confidence map for the fast bilateral solver.

### 4.2 Training Details

**Representation Learning.** We trained the depth representation encoder that takes on the architectural structure of ResNet-50 (He et al., 2016) with an additional fully connected layer as suggested in Chen et al. (2020a;b). Following the experimental setup in Chen et al. (2020b), we set $\tau = 0.07$,

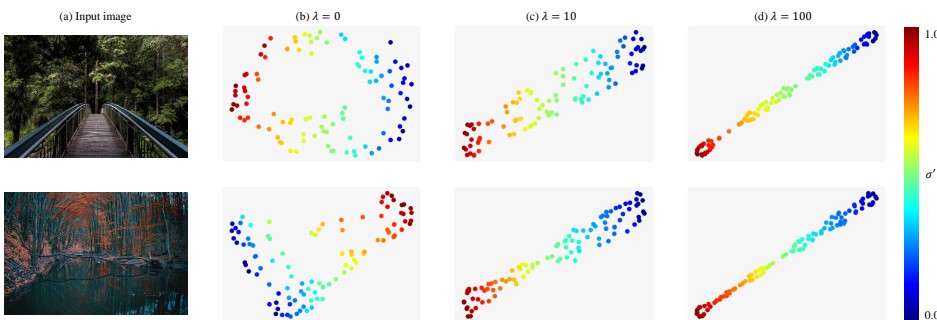

Figure 4: t-SNE visualization of representation space, color coded by the strength of depth enhancement effect. (b) Features are widely dispersed w/o regularization. (c), (d) As $\lambda$ increases, features at opposite extremes of depth perception are located far away from each other.

$K = 65536$ and momentum to 0.9. The learning rate was initialized to 0.03 and adjusted with a cosine annealing scheduler. The network was trained on 8 V100 GPUs for 2000 epochs with a batch size of 32. We used DIV2K train set (Agustsson & Timofte, 2017), Flickr2K (Lim et al., 2017) and Waterloo Exploration Database (WED) (Ma et al., 2017) to obtain a total of 8194 high quality images. In unsupervised representation learning, it is important to show a variety of strengthening effects to train a rich representation space. Therefore, we randomly sampled each parameter of depth enhancement model uniformly from the following range: $\alpha \in \mathbb{R}[0.15, 0.3]$, $\eta \in \mathbb{R}[0.7, 0.9]$, $\delta_{contrast}, \delta_{blur} \in \mathbb{R}[0.35, 0.5]$ and $s \in \mathbb{R}[11, 33.8]$.

**DEEAPR Training.** The DEEAPR network was trained on 8 V100 GPUs for 5000 epochs with a batch size of 32 using the same dataset. We used the representation encoder trained at $\lambda = 10$ and fixed its weights during training of the depth enhancer. The learning rate was initialized to 0.005 then adjusted with a cosine annealing scheduler. The parameters of depth enhancement model were fixed for consistency: $\alpha = 0.225$, $\eta = 0.8$, $\delta_{contrast} = \delta_{blur} = 0.425$ and $s = 22.4$.

## 5 RESULTS

### 5.1 ANALYZING REPRESENTATION SPACE

In this section, we analyze the role of the proposed triplet margin loss, $\mathcal{L}_{reg}$, defined in equation 1 as a regularization function in learning the depth representative feature space independent of content. The effectiveness of $\mathcal{L}_{reg}$ is observed by training the depth representation encoder (Sec. 3.1) at various levels of $\lambda \in \{0, 10, 100\}$, then performing t-SNE analysis on the global depth representation features, $f_{global}$, at a specified value of $\lambda$. Specifically, we randomly selected 10 images from the DIV2K validation set and generated 100 different depth-enhanced versions for each image by applying the depth enhancement model (Sec. 4.1). For visualization purposes, we defined a variable $\sigma'$ to control the numerous parameters that regulate the strength of depth enhancement with a single parameter. That is, the five parameters in the depth enhancer, $\alpha, \eta, \delta_{contrast}, \delta_{blur}$ and $s$, were linearly interpolated between their weakest and strongest values according to $\sigma'$ (e.g., $\eta' = \eta_{\min}\sigma' + \eta_{\max}(1 - \sigma')$), where $\sigma'$ was uniformly sampled from $\mathbb{R}[0, 1]$.

Fig. 4 shows the distribution pattern of encoded features in the representation space, where each feature point is color coded by the regulation parameter $\sigma'$ (i.e., strength of depth enhancement). At $\lambda = 0$, the distance between points that correspond to similar values of $\sigma'$ is similar to the distance between points that correspond to extreme values of $\sigma'$, resulting in a widely dispersed scattered plot. In contrast, as $\lambda$ increases, the points that correspond to extreme values are pushed away from each other while those with similar values are pulled towards each other, resulting in a thin band of gradation with extreme values at distant ends.

### 5.2 DEPTH PERCEPTION ENHANCEMENT

We expect the DEEAPR network to enhance the depth perception of a given image when $\sigma = 1$ and show smooth transition of depth enhancement effect within the range of $\mathbb{R}[0, 1]$. Indeed, Fig. 5

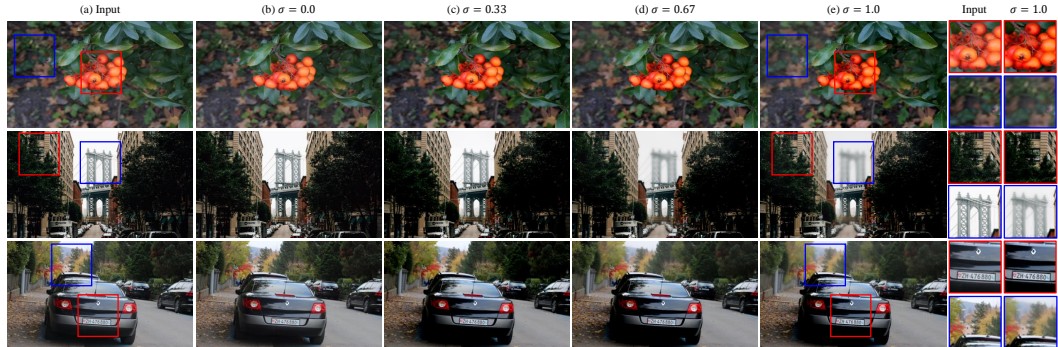

Figure 5: Output of DEEAPR when $\sigma \in [0, 1]$ showing gradual change of the depth enhancement effect. Results are best viewed electronically with zoom. See supplemental for additional examples.

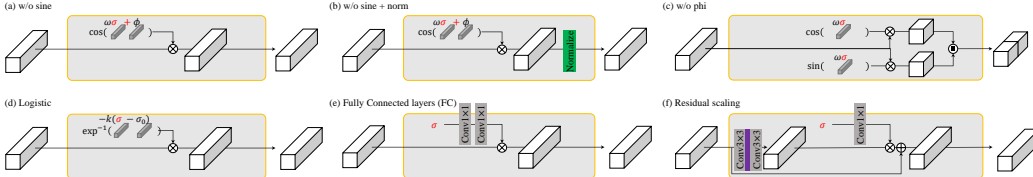

Figure 6: Six variants of PFRB used in the ablation study. Purple box indicates GELU activation.

shows that DEEAPR enhances depth perception of the input (column (a)) by simulating contrast and blur effect when $\sigma = 1$ (column (e)). Contrast enhancement of foreground objects (red boxes) strengthens volumetric perception, while the background (blue boxes) becomes blurry and hazy, making the foreground objects further stand out. Furthermore, DEEAPR not only produces plausible outputs when $\sigma \in \{0, 1\}$, but also exhibits smooth transition of effects in range $\mathbb{R}[0, 1]$. More specifically, close objects (the berries in the first row, trees and buildings in the second row and the car in the third row) show gradual enhancement of contrast as $\sigma \to 1$ and far objects (leaves in the first row, the bridge in the second row and the trees in the third row) become blurry and have decreased contrast as $\sigma \to 1$, producing perceptual depth enhancement effects.

## 5.3 ABLATION STUDY

In this section, we study the importance of the PFRB design in producing the controllable depth enhancement effect. To this end, we designed six additional controllable module variants and trained the DEEAPR network using the same pretrained representation encoder and condition as described in Sec. 3.1 and 4.2, resp. (see Fig. 6 for illustration of the six PFRB design variants, (a) to (f)). (a) To observe the effectiveness of modulating depth representative feature using the complementary sinusoidal functions, we *ablated the sine* function in the modulator. (b) To observe the importance of norm preservation via the Pythagorean trigonometric identity, we embedded an additional *normalization* term to the cosine function in (a). (c) The importance of modulating the control parameter $\sigma$ by affine transformation was sought by *ablating* $\phi$. We further explored alternatives to modulating the control parameter by using (d) a *logistic function* with trainable midpoint $\sigma_0$ and curve steepness $k$, (e) *fully connected layers* and (f) *residual scaling*. (e) and (f) were motivated from the controllable component of CFSNet (Wang et al., 2019a) and CResMD (He et al., 2020a), resp.

To study the effectiveness of each controllable module, we compared the variability of the output according to the control parameter $\sigma$. For each image in the DIV2K validation set, we first obtained the output at various levels of $\sigma$ (e.g., $\sigma = 0.0, 0.1, \cdots, 0.9, 1.0$). Next, we encoded each output with the pretrained depth representation encoder commonly used across all six variants and the proposed DEEAPR model (referred to as the baseline) and measured the cosine similarities of encoded features with respect to the encoded feature of the output produced at $\sigma = 0$.

The cosine similarities of the encoded features of the output is illustrated in Fig. 7(a). The proposed DEEAPR model (blue) demonstrates the second largest variability according to $\sigma$ with a smoothly decreasing cosine similarity curve as $\sigma$ increases. While Fig. 6(f) *residual scaling* (pink) showed

the most discrepancy between $\sigma = 0$ and $1$ in the cosine similarity curve, visualization of the output results in Fig. 7(c) revealed that *residual scaling* failed to reproduce the input at $\sigma = 0$ and to produce a depth enhanced image at $\sigma > 0$. In addition, the cosine similarity plot revealed that Fig. 6(a) *w/o sine* (orange) and Fig. 6(d) *logistic function* (purple) failed to produce a variable output, which was further confirmed with no visible changes in the image domain. While Fig. 6(c) *w/o phi* (red) was able to produce a variable output in the encoding space and image space (as shown in Fig. 7(a) and (c), resp.), it was less effective than the proposed model. Altogether, our study revealed that using *both* cosine and sine functions is crucial for modulating features obtained via representation learning.

## 5.4 COMPARISON TO OTHER CONTROLLABLE METHODS

In this section, we compare DEEAPR to other controllable methods. Specifically, we consider image interpolation, network interpolation (Wang et al., 2019b) and two other controllable methods: CFSNet (Wang et al., 2019a) and CResMD (He et al., 2020a).

For *image interpolation*, we obtained an image with smooth variation in depth by interpolating between the input image and the depth enhanced image. That is, suppose the DEEAPR network is trained at fixed $\sigma = 1$ such that the network *always* enhances the depth perception of the input. Then the image interpolated output with controllable parameter $\sigma' \in \mathbb{R}[0, 1]$ is defined as follows:

$$\mathbf{O}_{\text{img\_interp}}(\mathbf{I}; \sigma') = (1 - \sigma')\mathbf{I} + \sigma' f_\theta(\mathbf{I}, 1), \tag{5}$$

where $f_\theta(\mathbf{I}, \sigma)$ is the DEEAPR network with parameters $\theta$. Note that we use $\sigma'$ in this section to avoid confusion with $\sigma$, which is an input to the DEEAPR network.

For *network interpolation*, we interpolated network parameters associated with the input image and the depth enhanced image. The network affiliated with the input image was obtained by fine-tuning the trained network, $f_\theta(\mathbf{I}, 1)$, to learn identity mapping (i.e., to reproduce the input image at $\sigma = 1$). The network interpolated output was obtained by linearly interpolating the network parameters according to $\sigma'$ as follows:

$$\mathbf{O}_{\text{net\_interp}}(\mathbf{I}; \sigma') = f_{\theta'}(\mathbf{I}, 1), \text{ where } \theta' = (1 - \sigma')\theta_{\text{finetune}} + \sigma'\theta \tag{6}$$

For CFSNet and CResMD, we referred to the original papers for architecture and trained the networks under the same condition as DEEAPR.

Similar to Fig. 7(a), we show the correlation between encoded features of the output at various levels $\sigma' \in \mathbb{R}[0, 1]$ with respect to the output at $\sigma' = 0$ in Fig. 7(b) and the interpolated results at select values of $\sigma'$ in Fig. 7(c). The cosine similarity curve indicates that network interpolation (green) does not decrease monotonically as $\sigma' \to 1$ but forms a v-shaped curve, implying that this method does not offer a continuous strengthening effect in range $\mathbb{R}[0, 1]$. Again, the proposed method (blue) shows the largest variability among other controllable methods, as well as visually pleasing output.

## 6 CONCLUSION

In this paper, we propose a controllable neural network that enhances the depth perception of the given image by modulating the depth representative feature obtained from the pretrained depth representation encoder. To train the content-independent depth representative feature space, we introduce a hard negative sample that shares the same content as the query and the positive sample but differs in depth perception. The representation encoder is regularized with a triplet margin loss, which plays a crucial role in disentangling style (i.e., depth perception) from content. The sinusoidal operations in the parametric feature rotation block (PFRB) is key to generating controllable depth enhancement effect as they permit smooth variation of representative features residing along the unit hypersphere.

We want to emphasize that our strategy is not limited to controllable depth enhancement but can be extended to other image-to-image translation scenarios; for example, a controllable image restoration model can be formulated by training the representation encoder to learn a content-independent representation of degradation quality and the enhancer to restore the low quality input image. In addition, our unsupervised visual representation learning strategy is effective even when the *style* is difficult to be measured or quantified as we focus on learning the style space manifold. Thus, our DEEAPR framework is applicable as a general purpose image-to-image translation method with single-parameter controllability.

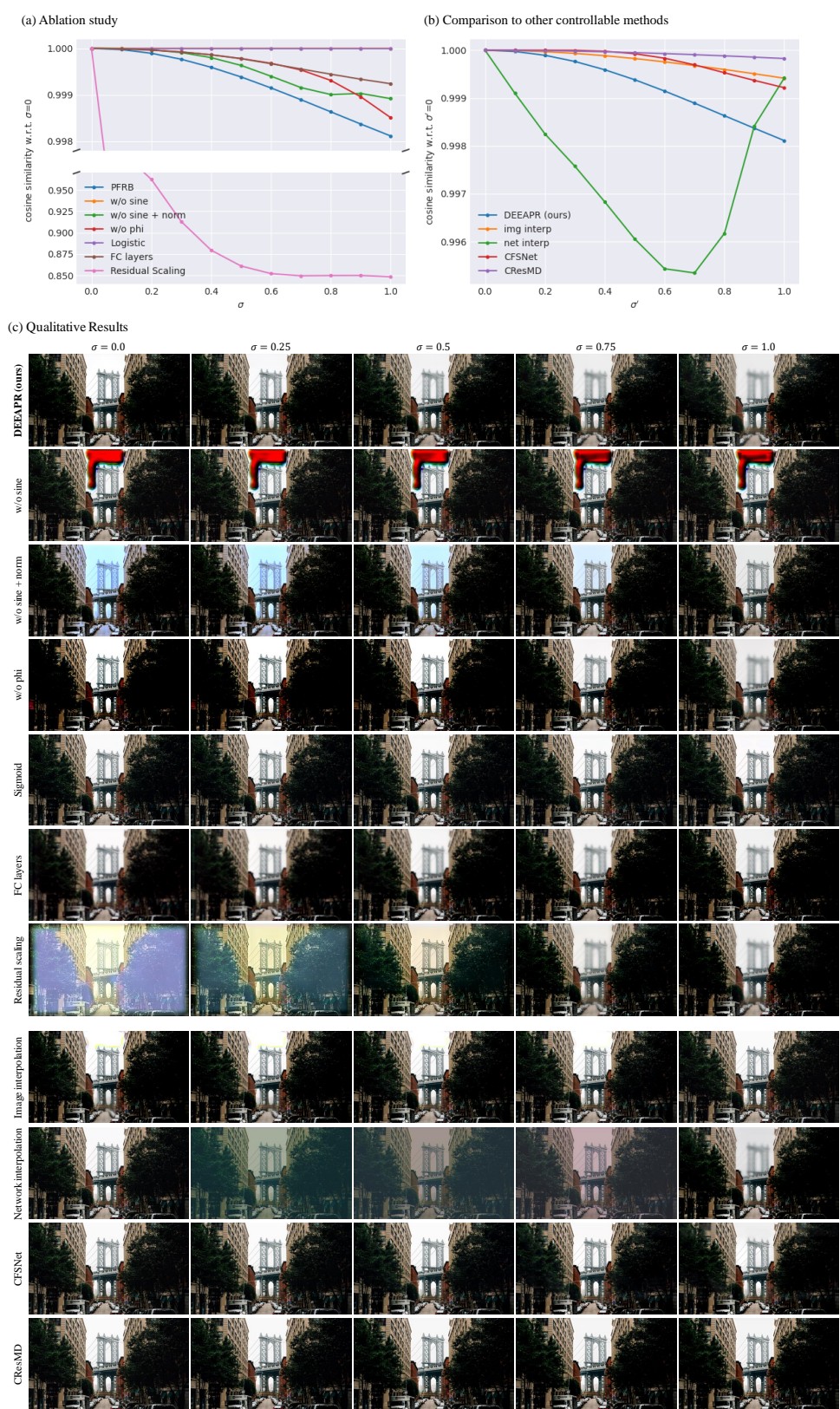

Figure 7: Cosine similarities of encoded features of output at various levels of $\sigma$ with respect to the encoded feature of output at $\sigma = 0$, comparing to (a) different PFRB designs and (b) other controllable methods. (c) Qualitative results of DEEAPR and other methods. All networks (except for image interpolation and network interpolation) are trained with $\sigma \in \{0, 1\}$ then evaluated in range $\sigma \in \mathbb{R}[0, 1]$. Best viewed electronically with zoom. See supplemental for additional examples.

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
