# OpenReview forum: "DEEAPR: Controllable Depth Enhancement via Adaptive Parametric Feature Rotation"
_ICLR.cc/2023/Conference — Submitted to ICLR 2023_

### Official Review · Reviewer_MFV5 · 2022-10-16

**Confidence:** 3
**Clarity, Quality, Novelty And Reproducibility:** This paper is overall easy to follow.…
**Correctness:** 3
**Technical Novelty And Significance:** 2
**Empirical Novelty And Significance:** 2
**Recommendation:** 3

**Strength And Weaknesses:**

Strength
- This paper proposes a strategy to learn the visual representation space of style disentangled from image contents.
- A controllable neural network with a single control parameter is proposed to enhance the depth perception of the input image.
- A parametric feature rotation block (PFRB) is introduced to enable continuous modulation of feature representation.

Weaknesses
- It is unclear to me the advantage of this method over existing methods utilizing explicit depth information. In fact, the ground-truth depth-enhanced images are just generated using the estimated depth maps of the input image. Also, the controllable factor (i.e., sigma) can be utilized in methods using explicit depth as well.
- Lack of comparison with other depth enhancement methods.
- There is no numerical result. I understand that quantitative evaluation is difficult, but a user study might be the least that can be provided.
- The qualitative results are not visually pleasing and unrealistic. There are many obvious errors in the generated images, e.g., in Fig.7(c) regions with different depths have the same blur effects, and regions with similar depths have different blur effects.


**Summary Of The Paper:**

This paper proposes a method for enhancing the depth perception of an image. The proposed method first embeds content-independent depth perception of a scene using the visual representation learning technique, and then trains a controllable depth enhancer network based on a parametric feature rotation block (PFRB). Some visual results are shown to verify the proposed method.


**Summary Of The Review:**

Given that the idea is not very new, the reported results are not visually pleasing, and the evaluation is not sufficient, I would like to give a reject rating at the current stage.

---

### Official Review · Reviewer_mEsF · 2022-10-21

**Confidence:** 4
**Correctness:** 2
**Technical Novelty And Significance:** 2
**Empirical Novelty And Significance:** 2
**Recommendation:** 3

**Clarity, Quality, Novelty And Reproducibility:**

The paper will need a clear clarification of its key contributions and a clear explanation of its motivations. The paper provides a clear description of network details for reproduction but it is not clear whether the code & data will be released to the public.

**Strength And Weaknesses:**

Comments.

The paper lists three contributions. The first one is a new method to learn style disentangled visual representation space of image contents. This mostly relates to visual representation learning, but the difference between the proposed method and existing methods is not explained. Sec. 2.1 mentions that it is similar to (Chen et al., 2020b) and is motivated by DASR (Wang et al., 2021), but it is not clear what the exact similarities and differences are. Hence, it is very hard to evaluate this contribution.

Besides, it is not clear how to define the "content-independent depth perception of a scene" and why "content-independent" relates to this task. How can the combination of (Hadsell et al., 2006) and (Balntas et al., 2016) embed the content-independent depth perception of a scene onto a representation space, and why this bridges the image space and the control parameter axis? Why choose these two methods? How to define the control parameter axis? What is the "direction" of depth enhancement?

The second contribution is a controllable neural network that enhances the depth perception of an image with a single control parameter. Sec. 2.2 says the main differences to existing methods are that DEEAPR uses a single parameter and does not rely on explicit depth information. However, it seems that (Dutta et al., 20) also do not rely on explicit depth information. So the key difference is about using a single parameter. It is not explained why the authors propose to use a single control parameter. Is a single control parameter enough and why?


The third contribution is the parametric feature rotation block (PFRB) for continuous modulation of feature representation. Its motivation and novelty are not explained clearly. For example, (Wang et al., 2019a) have a tuning network to modulate the feature of the main branch. How does the proposed DEEAPR differ from this?

Almost all methods included for discussion and comparisons are published before/in 2021. More recent works should be included.

**Summary Of The Paper:**

This paper proposes the DEpth Enhancement via Adaptive Parametric feature Rotation (DEEAPR) method to modulate depth information with a single control parameter. They first use visual representation learning to embed content-independent depth perception of a scene, then train a controllable depth enhancement network with a novel modulator (parametric feature rotation block, PFRB), which is for continuous modulation of a representative feature. They verify the effectiveness of proposed components and the whole method via ablated studies and comparisons.

**Summary Of The Review:**

The paper lists three contributions but I find them not convincing. This reflects in how they explain the motivations and discuss their differences with existing methods. Hence, at this point, I am on the negative side.

---

### Official Review · Reviewer_3SWe · 2022-10-25

**Confidence:** 5
**Correctness:** 3
**Technical Novelty And Significance:** 3
**Empirical Novelty And Significance:** 2
**Recommendation:** 3

**Clarity, Quality, Novelty And Reproducibility:**

### Clarity
The term *depth perception* used throughout the paper is vague and really is majorly DoF effect + depth-dependent haze effect at its core given the authors' results and depth-based model. The authors could just clear it upfront.
### Novelty
Network architecture and the proposed PFRB module seem novel
### Reproducibility
It may be difficult to reproduce the authors' results if the authors don't open-source their code, since the architecture and training setup is novel.

**Strength And Weaknesses:**

### Strengths
1. Controllable network for different DoF effect levels (though seen in previous work).
2. Network architecture and PFRB module seem novel.
### Weakness
1. Given Bokeh's image formation model and the DoF effect's strength are clearly defined, I have doubt whether learning a latent space for rendering is necessary.
2. The term "depth perception" is ambiguous, but rather DoF is the good latent space that models the background/foreground blur/sharpness contrast.
4. Lack of results to show the robustness of the proposed network. Only very few images (I counted 4 throughout the paper and supplementary material) are used for visual evaluation.
5. The shown results image has unnatural artifacts. For example, the bridge image, different portions of the bridge at the same depth are exhibiting different levels of blur.
5. The depth-based contrast model $O_{contrast}$ in section 4.1, though already redefined from the haze/fog model from prior, is still heavily haze-oriented. For general photos taken under good weather conditions, the proposed method may introduce an unwanted or exaggerated hazy effect.  The plant image in the supplementary material proves this point: as $\sigma$ increases, the further away background (maybe ~50$cm$) is becoming hazy, whereas the depth-dependant haze effect should hardly be seen in the real world for an object about this distance.


**Summary Of The Paper:**

The authors proposed a network-based approach to control the depth-of-field (DoF) effect of photographs post-capture. While there are many previous works in artificial Bokeh generation, the authors claimed their novelties lie in 1) learned representation space for the DoF effect, 2) controllable network for the strength of the DoF effect, and 3) a new network block called PFRB that can preserve the norm when modulating the features.

**Summary Of The Review:**

While I appreciate the effort the authors carry out in designing the network and module for the specific problem of post-capture DoF effect rendering which is very useful in computational photography and has a wide-range impact in real life.

My biggest concern is whether the authors' fundamental motivation, that we need a learnable latent representation space for the DoF effect, is valid.

Usually learning an implicit latent space for image manipulation or enhancement is crucial when the problem can not be easily and explicitly defined by physics-based models or the underlying physical model is too complex or ill-posed  (i.e. manipulating the age of the person in a given image, image style transfer, etc.). However, in the case of DoF or Bokeh effect rendering, the problem is very well and explicitly defined. The DoF is clearly defined as a function of the depth ($u$), camera focal length ($f$), aperture size or F-number ($N$) and circle-of-confusion ($\mu$), $DoF = \frac{2 \mu^2 N C}{f^2}$. Shall or deep DoF (or "depth perception" per this paper) can be nicely controlled and rendered with the camera parameter factored into $O(x)$, defined in equation 4.

---

### Official Review · Reviewer_4wXg · 2022-10-28

**Confidence:** 4
**Clarity, Quality, Novelty And Reproducibility:** The quality of the paper is low.
**Correctness:** 2
**Technical Novelty And Significance:** 2
**Empirical Novelty And Significance:** 1
**Recommendation:** 1

**Strength And Weaknesses:**

Strength:
+ The output image can be controlled by adjusting the input hyperparameters

Weakness:
- The paper is poorly written and extremely hard to follow. The countless grammatical errors, notions used without any references,  and bad paper organization make it almost impossible for readers to understand the paper content. The following listed ones are just a few: What is query, positive sample and hard negative sample mean (Sec. 3.1 and Fig.3)? What is depth representation means? What's the difference between global/local depth representative feature (Sec. 3.2)? How you can rotate the parametric features in PFRB?
- The technical contribution is marginal. I didn't find significant difference between images of different \sigma.


**Summary Of The Paper:**

This paper proposed a novel method for depth enhancement for single image. They used a single hyperparameters to control the synthetic defocus blurriness.

**Summary Of The Review:**

In summary, even though the paper shows that they can achieve depth enhancement effect in a single image by varying value of a hyperparameter, there's still a very large space for this paper to improve w.r.t. writing and performance.

---

### Decision · Program_Chairs · 2023-01-20

**Decision:**

Reject

**Justification For Why Not Higher Score:**

There are significant concerns regarding the clarity, lack for empirical results, and more importantly, the requirement (and robustness) of a learning-based method. The authors did not respond to the concerns raised.

**Justification For Why Not Lower Score:**

N/A

**Metareview: Summary, Strengths And Weaknesses:**

This work presents a learning-based approach for obtaining a depth of field effect in images. However, there are significant concerns regarding the clarity, lack for empirical results, and more importantly, the requirement (and robustness) of a learning-based method (i.e. why does one need this approach for this task given a predicted depth and analytical equations can be used to obtain similar outputs). The reviewers unanimously recommend rejection, and the authors did not respond to the concerns raised.